# Diagnosis of Myasthenia Gravis

**DOI:** 10.3390/jcm10081736

**Published:** 2021-04-16

**Authors:** Rossen T. Rousseff

**Affiliations:** 1Department of Neurology, Ibn-Sina Hospital, Sabah Health Area,, Kuwait City 13115, Kuwait; emg.doctor@hotmail.com; Tel.: +359-878-417-412; 2Science and Research Institute, Medical University of Pleven, 5800 Pleven, Bulgaria

**Keywords:** myasthenia, diagnosis, anti-acetylcholine-receptor, anti- muscle-specific tyrosine kinase (MuSK), repetitive nerve stimulation, neuromuscular jitter

## Abstract

The diagnosis of autoimmune Myasthenia Gravis (MG) remains clinical and rests on the history and physical findings of fatigable, fluctuating muscle weakness in a specific distribution. Ancillary bedside tests and laboratory methods help confirm the synaptic disorder, define its type and severity, classify MG according to the causative antibodies, and assess the effect of treatment objectively. We present an update on the tests used in the diagnosis and follow-up of MG and the suggested approach for their application.

## 1. Introduction

Myasthenia Gravis (MG) is one of the best understood human autoimmune diseases. The pathogenic autoantibodies against structures of the neuromuscular junction can be routinely identified in the majority of patients [1,2]. The pathophysiology of impaired neuromuscular transmission is studied in detail, and several tests are readily available to assess the synaptic disorder [3,4]. Some techniques have been perfected and have become more accessible over the years, while others tend to be neglected [5,6]. While a standard approach has been formulated in a number of texts, regional and other disparities in the access to different tests exist so that the investigation tactic may differ according to the circumstances. We present a review of recent experts’ opinions that may help clinicians approach the rewarding task of MG diagnosis parsimoniously.

## 2. Neuromuscular Transmission

The axonal action potential, reaching the terminal branches, depolarizes them and opens the voltage-gated presynaptic calcium channels. The influx of calcium triggers acetylcholine (ACh) release from the immediate store of quanta into the synaptic cleft. The ACh diffuses to the postsynaptic membrane and interacts with the ACh receptors (AChR) on its folds’ crests. As a result, postsynaptic membrane depolarization develops and reaches the Na-channels in the postsynaptic folds’ depth. The resulting end-plate potential (EPP) spreads to the sarcolemma and generates a muscle fiber action potential. In turn, it starts the cascade of events leading to muscle fiber contraction. The ACh is hydrolyzed by the acetylcholinesterase (AChE) into its constituents, that undergo reuptake by the terminal and resynthesis into ACh.

The excessive amount of ACh quanta released and the specific organization of the postsynaptic membrane ensure an EPP much exceeding the depolarization threshold of the sarcolemma; this is the “safety factor” of neuromuscular transmission that ensures prolonged repetitive muscle fiber contraction. The AChR clustering and other critical features of the end-plate are regulated by the release of the protein agrin from the terminal. Agrin activates the enzyme muscle-specific tyrosine kinase (MuSK) in complex with low-density lipoprotein receptor-related protein 4 (LRP4). Other features of the normal synaptic structure and function are outside our very brief synopsis; extensive reviews are available on the subject [7].

## 3. Antibodies in Diagnosis of Myasthenia Gravis

Detecting established pathogenic antibodies against some synaptic molecules in a patient with the typical clinical features is virtually diagnostic of MG and helps define the disease subtypes [1,5,6]. The autoantibodies to the AChR (AChR-Ab) were identified and studied first [1]; their absence classified some patients into a “seronegative MG” group. Research over this cohort discovered the MuSK-Ab and, recently, the LRP4-Ab as causative in a smaller number of MG patients [8,9]. Assays are commercially available for detection of AChR-Ab, MuSK-Ab, and LRP4-Ab and are being perfected in different aspects, while other techniques are still restricted to “in-house” use at leading research institutions [2,5]. While not directly pathogenic, several other autoantibodies prove useful in assessing thymoma-MG and late-onset MG [2,6]. We review the methods, the indications, and the diagnostic significance of autoantibody tests concisely.

### 3.1. Anti-Acetylcholine Receptor Antibodies

In a patient with clinical features of MG, serum testing for AChR-Ab is the first recommended step in diagnosis by most recent guidelines and expert opinions [2,5,6,10].

The most widely studied are the binding AChR-Ab, using a radioimmunoprecipitation assay (RIPA) [1,2]. The highly selective AChR-agonist alpha-bungarotoxin, labelled with ^125^I, fixes to AChR extracted from muscle cell lines. The complex is then incubated with the patient’s serum and precipitates with circulating serum AChR-Ab. Measuring the radioactivity of the precipitate assesses the quantitative titer of the antibody. The test has a very high specificity, so the detection of elevated titers in the appropriate clinical setting is diagnostic of AChR-myasthenia, and further testing may not be necessary [5,6,10]. A degree of diagnostic suspicion is appropriate regarding the rare cases of antibody-positive subjects with other disorders [11] or human and technical error [12]. The observation that some MG patients have increased titer years before the clinical onset (as retrospectively found on preserved sera) may explain some of the apparent false-positive results [13].The RIPA for binding Ab is abnormal in up to 85% of adult patients with generalized MG, but only about 50% of patients with ocular MG [2,5,6,14]. Detection of modulating and blocking AChR-Ab through RIPA is commercially available but adds little diagnostic value [5], although others recently emphasized their significance [15]. In selected seronegative cases, the test may be repeated within 6–12 months, as delayed seroconversion is well-described, although rare [16].

The enzyme-linked immunosorbent assay (ELISA) for binding Ab is also available, relying on standard equipment and avoiding work with radioactivity; however, the prevailing opinion seems to be that ELISA is less specific and sensitive than RIPA [2,5,6]. Another alternative to radiolabeling may be the fluorescence immunoprecipitation assay (FIPA), which is also under study for MuSK-Ab detection [2,6]. Recently proposed are rapid tests for AChR-Ab detection by modified ELISA with AChR fixed on surfaces (“immunostick”, “dot-blot” techniques) [17,18]. They are a promising development with reported high sensitivity and specificity that needs further validation [2].

The cell-based assay (CBA) is based on the development of cell lines expressing the AChR and transfected with rapsyn, which stimulates clustering of the receptors. The cells are incubated with the patient serum, and immunofluorescence is applied to detect the autoantibody, in this case, AChR-Ab against clustered receptors. The technique detects autoantibodies to clustered AChR in about 15% of RIPA-seronegative MG patients [19]. The CBA is an in-house qualitative method, valuable in research, but highly complex and difficult to introduce as a routine. It would miss some patients, positive by RIPA [2,16]. Some centers offer the CBA study as part of a panel for seronegative myasthenia.

### 3.2. Antibodies to MuSK

Autoantibodies to MuSK, the postsynaptic receptor for agrin, were identified in 2001 as causative in a prevailing percentage of the patients with “seronegative MG” [8]. Soon after, the collective efforts of different centers defined MuSK-MG as a specific disease subtype that differs significantly from the AChR-MG review in [20,21]. MuSK-Ab are found in about 6–8% of all MG cases [2,5]. Detection of MuSK-Ab is commercially available, mainly by RIPA, using the directly radiolabeled antigen, ^125^I-MuSK (extracellular domain) [20]. The study is nearly 100% specific; the sensitivity is less clear, as the proportion of MuSK-MG patients differs among study groups according to their origin, respectively, to certain genetic predispositions [20,21]. ELISA kits for MuSK-Ab detection are available but not widely used [2]. CBA detected MuSK-Ab in patients, RIPA-negative both for AChR-Ab and MuSK-Ab [20]. As some of them harbored IgM MuSK-Ab of unclear significance, an IgG-specific CBA for MuSK-MG was developed [22].

### 3.3. Antibodies to Anti-LRP4

The lipoprotein receptor-related protein 4 (LRP-4) mediates agrin signaling to MuSK. LRP4-Ab were confirmed as causative in the last decade only; their presence varies widely and depends on the geographic area [9,23]. LRP4-Ab may be pathogenic in about 2% of MG patients [2,23]. They are identified by CBA, RIPA, and ELISA; the methods are still being validated [2,6]. Furthermore, LRP4-Ab are present in up to 23% of amyotrophic lateral sclerosis (ALS) patients, a percentage of patients with other neuroimmune disorders, and up to 20% of subjects with MuSK-MG (as defined by serological and clinical criteria) [2]. Thus, the detection of anti-LRP4-Ab is not a straightforward diagnostic, unlike that of AChR-Ab or MuSK-Ab; it should be interpreted strictly within the clinical context [6,24].

### 3.4. Double-Seropositive Myasthenia Gravis

MuSK-Ab are increasingly investigated in AChR-Ab positive patients, who are treatment-resistant or develop clinical features less typical of AChR-myasthenia [21]. Thus, a group of double-seropositive MG patients was identified that seemed to be more frequent in East Asian populations. Some authors suggest that these double-seropositive subjects resembled more MuSK-MG cohorts [25]. The group of “double-seropositive” MG may involve up to 12.5% of all patients [2]. Recently, “triple-seropositive” patients were also observed [26]; this underlines the significance of an integrative approach to MG classification and management [27].

## 4. Pharmacologic Tests

Disordered neuromuscular transmission (NMT), due to a decreased number of functioning AChR, may be improved by inhibiting the AChE enzyme; it provides larger amounts of ACh to act on the remaining AChR over an extended period [28]. For diagnostic purposes, an AChE-inhibitor with immediate-onset and short-lasting action would be optimal so that the clinical effects relate closely with its application. Edrophonium chloride (Tensilon, Enlon) effects start within 30 s and last about 5 min after intravenous injection; it was used in MG diagnosis since the 1950s [29]. It may cause a dramatic recovery, especially of oculomotor functions (Figure 1).

Edrophonium evokes cholinergic side effects, such as salivation, sweating, nausea, and fasciculations. The risk of serious side effects is low (less than 0.2%) [31] but not negligible and includes bradyarrhythmias to asystole and cardiac arrest, bronchospasm, seizures, and transient ischemic attack(TIA) [31,32]. The potential liability issues have limited the use of the study. It should be done by an expert [15] in a setting allowing to react to such effects, monitoring the heart rate and blood pressure [10,15]. A total of 0.6 mg atropine in a separate syringe should be available. Individual sensitivity to edrophonium differs, so the medication is given intravenously in fractionated doses up to the maximum of 10 mg (2 mg + 3 mg + 5 mg, 2 mg + 8 mg within 45 s) [5,10]. Some authors advise placebo application, but the side effects of edrophonium nearly always “unmask” the active substance and may provoke a positive placebo result [6]. To avoid bias, one should select a measurable sign (e.g., eyelid fissure height) and document the objective effect.

In MuSK-MG, edrophonium may worsen patients’ weakness [33], so in clinical suspicion of MuSK-MG, the test is best not performed.

An alternative to edrophonium is the slower, longer-actin neostigmine, which is introduced intramuscular (IM) The improvement is expected to start within 5–15 min, becomes most apparent later, and may last hours. The selection of objective endpoints in the neostigmine test seems to be less reliable [10]. This test may be preferable to edrophonium in younger children who cannot cooperate fully [5,10].

The pharmacological tests’ sensitivity and specificity have rarely been reported in a methodologically sound way [34]. Some authors claim that the neostigmine test is more sensitive and specific than antibody testing and electrophysiology study [35], but this seems to contradict the prevailing experience [10,36].

In summary, pharmacological tests are useful, especially in ocular myasthenia [5], but logistical problems related to safety, drug availability, and some ambiguity in the choice of the endpoints have limited their use. In the USA, edrophonium has been discontinued since 2018 [37]. The Italian recommendations mention that neostigmine may be used as a third-line test in seronegative patients with normal electrodiagnostic results [38]. The Guidelines of the Association of British Neurologists state that “…The edrophonium test may be difficult to interpret. Conditions that mimic myasthenia may produce a positive result, and there are potential cardiac complications”, [39]; to our knowledge, it has not been practiced in the UK in the last years.

On the other hand, in many areas, access to reliable serological or electrodiagnostic study is limited or even impossible, not the least for monetary reasons. In such circumstance, the cheap and available Neostigmine test may still be an important diagnostic tool in MG.

## 5. The Ice-Pack Test

Heat was noted to worsen an MG-patient’s fatigue and weakness, while cold, on the contrary improved them [40], which gave the idea for a simple local cooling test: application of an ice-pack over a symptomatic eye for 2–5 min was found to reduce ptosis and ophthalmoparesis [41]. The test is used mostly in ocular MG and seems more popular among neuro-ophthalmologists; some authors report sensitivities equal to that of single-fiber electromyography (SFEMG) in patients presenting with ptosis and a very high negative predictive value [42]. However, others note much lower specificity at about 25% [36]. The criterion for abnormality (improvement of ptosis by at least 2 mm) seems somewhat arbitrary, as improvement of this magnitude was seen even with the application of heat rather than cold (although, statistically, the cold pack was more effective in the resolution of ptosis) [43]. Some of the suggested diagnostic algorithms or flowcharts do not include the ice pack-test [6]. Still, this safe and straightforward bedside investigation, with the contemporary possibilities to film the results, has its place in practice, especially when the effect is unequivocal.

## 6. Electrodiagnostic Studies

The specialized electrodiagnostic techniques in MG are the repetitive nerve stimulation study (RNS) and single-fiber electromyography (SFEMG). Routine needle electromyogroaphy (EMG) and nerve conduction studies are performed to exclude alternative diagnosis or to confirm a concomitant disorder. In routine needle EMG of a weak myasthenic muscle, instability of the motor unit potential (MUP) with consecutive discharges may be noted by a seasoned neurophysiologist but remains dependent on subjective interpretation [5,10].

### 6.1. Repetitive Nerve Stimulation

Muscle responses to supramaximal repetitive nerve stimulation of different frequency, duration, and pattern are studied in synaptic (including the presynaptic) pathology and in some channelopathies for assessment of neuromuscular blocks, etc. Below, we focus on the changes in MG. Jolly (1893) initially observed them, using mechanical equipment to record the decrease of consecutive responses cited after [44]; the decline of electrically registered muscle potentials was documented in MG Harvey and Masland, 1941, after [44].

The compound muscle action potential (CMAP), in response to a supramaximal electric stimulus over the corresponding nerve, represents the sum of the action potentials (APs) of all muscle fibers within that muscle. In healthy muscle, all fibers respond to repeated stimulation in the physiological range of discharge rates. This stability is ensured by the “safety factor”: the amount of ACh released is larger than the required release, and the structures of the postsynaptic membranes are arranged in such a way that the end-plate potential (EPP) generated by the nerve impulse is much higher than the threshold values of the muscle fiber membrane [45]. Successive stimuli at a low rate produce smaller EPP, as only ACh from the immediate store is released, but the EPP remains high enough to depolarize the sarcolemma. This and other adaptations facilitate neuromuscular transmission and synchronize muscle fibers’ discharges in time [7].

The “take away” knowledge from this brief synopsis is that, with low stimulation rates (below the physiological), usually 2 or 3 Hz, the successive CMAPs in a healthy muscle will be identical by amplitude and area. With higher stimulation rates in the physiological range (10–20 Hz) and after prolonged voluntary contraction at maximal effort, the adaptive changes lead to some increase of the amplitude of the CMAP, but its area remains constant. This phenomenon is known as pseudofacilitation [4,7]. A schematic representation of the events during the normal RNS test at a slow rate can be found in Figure 2.

In MG, the safety factor is decreased due to the loss of acetylcholine receptors, sodium channels, and the derangement of the normal postsynaptic structure [46]. As a result, at a low rate of stimulation, the physiological drop in EPP amplitudes reaches a stage at which the EPP remains below the threshold for muscle fiber activation. The muscle fiber remains inactive, i.e., blocking appears. With successive stimuli, a growing percentage of the muscle fibers will block. The CMAP decreases in amplitude and area with repeated stimuli; an abnormal decrement is observed. This drop-in amplitude is calculated as a percent of the initial CMAP and is maximal between the first and second responses but reaches a nadir at the fourth and fifth responses [4,7,10], e.g., a 35% lower amplitude will translate to “35% decrement observed” at the fourth response. The responses then stabilize and may even increase on behalf of the true facilitation mentioned. These processes are schematically illustrated in Figure 2. The “staircase” or “U” shape of the train of stimuli is considered characteristic of postsynaptic pathology (Figure 3 and Figure 4) [47,48].

The abnormality limit for decrement is 10% of the initial amplitude/area, but with technically perfect recording, any decrement of the typical pattern is suspicious and deserves clarification [49]; the original traces for any non-physiological changes (movement, baseline instability). A minor decrement in single muscles should not be overestimated as diagnostic. In some cases, a muscle with normal or borderline initial RNS reveals true decrement immediately after 1-min of maximal contraction [48].

Changes of RNS due to postsynaptic pathology are further characterized by the effects of tetanic muscle contraction. It may be achieved via high-frequency electrical stimulation (30–50 Hz), which is poorly tolerated; most laboratories use maximal voluntary contraction over a brief period (10–60 s) instead [4,48,49]. Thereafter, the accumulation of calcium in the terminal mobilizes the ACh stores and tends to compensate or even completely reverse the transmission failure seen at lower frequencies. Therefore, the CMAPs immediately after exertion demonstrate some true increase (post-tetanic or post-exercise facilitation). However, further stimulation reveals worsening of decrement (post-tetanic or post-exercise exhaustion lasting 2–5 min) by mechanisms that are still not completely clear [48,49]. These phenomena are illustrated in Figure 5. The full sequence of post-exercise events is not often seen in postsynaptic pathology but is very prominent and is an obligatory study in suspected presynaptic disorder [50]. In MG, the additional diagnostic value of post-exercise testing was assessed at a few percent only, and some authors suggest to skip this step, using the time for testing other nerve-muscle combinations instead [51].

The accuracy of RNS in different nerves/muscles has been amply studied for decades, with variable results easily explained by study design, spectrum, and incorporation bias, choice of muscle, and, not the least, by different normative values for decrement. In generalized MG, RNS of distal muscles is abnormal in 30–35% of patients, while in proximal muscles, it is abnormal 60–70% [52,53]. In ocular MG or ocular onset MG, the RNS of distal muscles is disappointing at about 10% sensitiviy, while in the facial muscles, it may reach 35–38% [48,49,50]. Studies on new-onset MG patients compared to real-life differential diagnostic groups may be most useful but are not numerous [34,52,53]. One should note as a standard feature the high specificity of RNS, which reaches about 97% in generalized MG and 94% in ocular MG [34], or even 100% when studying six pairs of muscles [54].

The muscle to study by RNS should ideally be clinically involved. Testing more muscles increases the diagnostic yield [52,53,54]. The choice of nerve/muscle is determined by a patient’s tolerance and technical factors, in addition to the expected sensitivity. The hand muscles are easier to test and are better tolerated, but less sensitive [10,52]. Proximal muscles (like deltoid) give a high yield but require adequate immobilization and excellent cooperation [10,49,55]; among them, the trapezoid seems easier to test [56]. The anconeus muscle at the elbow is rated among the most sensitive in ocular MG [54] and in oculobulbar MG in some studies [57]. In the facial muscles, the sensitivity is the highest, and they are instrumental in ocular MG [54], but technically, they are more demanding and prone to patient movement issues [48]. In bulbar onset MG, stimulation of the hypoglossal nerve while recording from the submental muscle complex [54] is recommended. approaches, which aren’t standard, emerge in particular settings, e.g., masseteric nerve stimulation via a monopolar needle in bulbar onset or phrenic nerve RNS in respiratory weakness [58,59].

A recent study questions the 10% limit of abnormality for the decrement; with a 7% cut-off, the sensitivity of RNS increases significantly, while the specificity is much less affected [60]. Previous reports have also noted decrements of 5–7% at low rate RNS as maximal in healthy controls [50]. However, introducing a stricter cut-off limit increases the technical requirements.

A decremental response may be seen in denervating/reinnervating muscles, particularly with ALS, but differs from the typical postsynaptic decrement in distribution and pattern [55]. In a large cohort of 85 ALS patients, none had a significant decrement in a facial muscle; besides, the typical “U” shape, with some reversal of decrement at the fifth and sixth responses, was not observed in ALS [49,55].

Besides the high specificity, advantages of RNS are the easy repeatability and the correlation of RNS changes with the severity of neuromuscular transmission defect [61]. Thus, it may be used to monitor treatment response. RNS abnormality in a peripheral muscle in ocular onset MG increases the risk of generalization, according to some authors [62,63]. In an emergency setting, RNS may distinguish a cholinergic crisis from a myasthenic crisis [48]. It is useful in seronegative myasthenia [5,6,10]. However, the study is not always well-tolerated, especially in facial and proximal muscles or in obese patients.

### 6.2. Neuromuscular Jitter Study

In the early 1960s, Ekstedt and Stalberg designed EMG electrodes able to record the potentials of single muscle fibers in situ (single-fibre EMG, SFEMG), intended mostly for the study of physiological fatigue [64]. With the ingenious research of Stalberg, Trontelj, and others, the single-fiber electromyography (SFEMG) developed into an essential tool in the research and diagnosis of neuromuscular disorders [65].

Recently, because of epidemiological restrictions, the disposable, smallest size concentric EMG needle (28–30 G) is used for jitter measurements, although it is clear that the potentials recorded are not always true single muscle fiber potentials [66]. The study performed with such an electrode should be called “jitter measurement with concentric needle electrode” or “concentric needle (CNE)jitter” [66,67].

Understanding of jitter measurement returns us to synaptic transmission mechanisms. The time from ACh release to EPP generation and further, to muscle fiber activation, normally fluctuates within tens of microseconds, depending on the random oscillations of the amount of ACh, the number of refractory receptors, and the sarcolemmal potential. Thus, even in a healthy muscle, there is a variation in the transmission time; a neuromuscular jitter. It is calculated according to a specific algorithm as the mean consecutive difference (MCD) in time between successive discharges. It is estimated for individual fibers/pair of fibers (MCD per pair) and as a mean for the studied muscle (MCD per study) [7,68]. The jitter may be measured for a pair of fibers, voluntarily activated by the same axon (volitional jitter). In this case, one fiber was used as a trigger. With the stimulated CNE jitter, one applied repetitive, low stimuli over the motor nerve that activated only several axons at a time (microstimulation). The examiner adjusted the recording needle to select fiber potentials, corresponding to specific criteria. A schematic explanation of the recording methods may be seen in Figure 6.

Examples of normal jitter are presented in Figure 7 and Figure 8.

In postsynaptic pathology, the EPP becomes even more variable on behalf of decreased receptor numbers and deranged synaptic morphology. It may reach the threshold later than normal, with greater differences in time between discharges, i.e., the jitter will be abnormally high. When the EPP does not reach the threshold at all, blocking will be registered [7,68]. Schematic presentation of jitter recordings is given in Figure 6. Examples of normal jitters are seen in Figure 7 and Figure 8, and abnormal jitter recordings are shown in Figure 9, Figure 10 and Figure 11.

The jitter study is the most sensitive indicator of impaired neuromuscular transmission [4,5,10]. Studies of RNS and jitter performed on the same muscle discovered that decrement never appears without a jitter abnormality; on the contrary, muscles with normal RNS may show prominent jitter changes, sometimes with up to 50% blocking [69,70]. The CNE jitter is abnormal in 98–100% in generalized MG and above 90% in ocular MG. These results have been replicated numerous times [7,14,34,52,53,68]. The most sensitive are the facial muscles around the eye, especially in ocular MG or ocular onset MG. Like in RNS, one should focus on a clinically involved muscle (the masseter or the tongue in bulbar onset, the cervical paraspinals in dropping head/axial onset, deltoid in generalized fatigue). After a routine EMG, making sure the muscle is not denervated or myopathic, the CNE-jitter will be measured and compared to established normative values per muscle. A normal jitter measurement in a weak muscle or in three optimally chosen muscles excludes MG [7] or at least carries a very strong negative predictive value of about 98% [71]. An important feature of jitter measurement is its sensitivity in seronegative myasthenia [72]. The degree of abnormality corresponds to the severity of the disease [61]. Like with RNS, some very mild changes should not be overemphasized [6,7].

While highest in sensitivity, jitter measurements are not specific and may be abnormal in other synaptic disorders, in denervation and reinnervation processes (poly- and mononeuropathies), some myopathies, and with motor neuron disorder (MND) in particular [7,52,53,68]. Recently, it has been stated that botulinum toxin’s cosmetic or therapeutic application should always be excluded by directed history [73]. In the clinical setting, the low specificity of SFEMG may be compensated, partially because not so many of the conditions with increased jitter appear as clinical MG-mimics [74]. The choice of muscle is also important in this aspect, e.g., in a patient with a history of radiculopathy C7, one would not rely on EDC muscle jitter study. Our collective has established that, in known diabetic neuropathy, jitter in the frontalis muscle is preserved, except in some very severe cases; in contrast, the EDC muscle jitter shows significant changes in most patients [75].

As the most sensitive technique in neuromuscular transmission disorders, jitter measurement may be decisive in seronegative MG. However, it is not readily available, even in the developed world; this is an expert study, much dependent on operator expertise and strict adherence to the technical criteria of recording and patient cooperation and tolerance [76].

In pediatric practice, a stimulated CNE jitter modification was introduced recently, which obviates the need for longer and deeper sedation in infants and young children (Stimulated Potential Analysis using Concentric Electrode (SPACE)). Complex signals consisting of multiple peaks are recorded from the orbicularis oculi muscle after repetitive stimulation of the zygomatic branch and analyzed visually and by peak-detection software; a detailed description of the technique is available in Pitt, 2017 [77].

## 7. Thymus Imaging

Thymoma is present in up to 15% of MG-patients, mostly in those with detectable AChR-Ab [6]. Several clinical and serological features are positively and negatively predictive for thymoma (e.g., in MuSK-MG, thymoma is seen in single cases only; in the presence of striational antibodies, thymoma is very likely), but none are absolutely reliable, so imaging of the thymus is advised in all confirmed or strongly-suspected MG patients [5,6,39]. The techniques for detecting and classifying thymic pathology and assessment for malignancy, etc., are outside the scope of the review. CT, MRI, PET, radioisotopic techniques all have a place [78], but for screening, CT seems preferred [5,10]. On the other hand, all patients with thymoma should be investigated for MG, as up to one-third of them develop the disease [79]; AChR-Ab may be positive in thymoma patients without clinical evidence of synaptic disorder, so electrodiagnosis may be necessary in dubious cases.

## 8. Diagnostic Approach

From the simple confirmation of an acquired postsynaptic disorder, the diagnosis of autoimmune MG has developed into the task of identifying the disease subtypes along several axes: the type of causative antibody, distribution of weakness, and age of onset, relation to thymic pathology [6,27,39]. Such classification proves very important in management, as those subtypes differ in pathophysiology and may require different treatments, e.g., thymectomy is recommended in AChR-MG of early onset but not in AChR-MG of late onset; anticholinesterases are less effective and may even worsen MuSK-MG, etc. The expected approval of biologicals targeting specific steps in pathophysiology (antibody production, complement action) stressed the need to distinguish the subtypes further.

Accordingly, the first step in a patient with the clinical features of myasthenia should be tested for AChR-Ab and MuSK-Ab. A positive result is diagnostic and there is no need of further confirmation, according to some authors [5,6,27]. The patient should be referred for screening of the thymus and thyroid function tests to exclude comorbidities. However, authoritative sources recently report specificities of 90–95%, with dozens of false positives in large series of patients tested for MG [15]. Such results may relate to the use of antibody testing as a screening rather than as a confirmatory test. This underlines the need for an index of suspicion in patients with less typical clinical features and a liberal use of additional tests and possible Ab retesting.

In a seronegative patient, electrodiagnostic tests are the next objective step in diagnosis. The RNS is less sensitive, but highly specific and more available. If the tests are negative, jitter measurement is performed. Electrodiagnostic studies should be directed at clinically involved muscles; it is wise to include as many muscles as possible (tolerated by the patient). If electrodiagnosis confirms postsynaptic disorder, the diagnosis is definite [6,10].

The pharmacological test (where available and when objectively unequivocal) may be sufficient for definite diagnosis of MG [5] and may be used as an alternative to electrodiagnosis as a second-line method, especially in ocular MG [14]. The use of a pharmacological test is also dependent on the availability of the medication.

The ice-pack test is recommended by some at this step as well, especially in ocular MG; again, many experts would not include it in the guidelines or suggested algorithms [6,38,39], but there are also proponents of its high accuracy [42].

If a patient is negative for all diagnostic procedures above and the clinical picture is of clearly fluctuating asymmetric ptosis and double vision, then ocular MG is probable; treatment with clinical observation and retesting should be considered, especially if the duration of disease at first tests was short. According to some UK guidelines [39], brain imaging may be appropriate to exclude some of the rare mimics due to structural brain disease. Thyroid finction tests are appropriate in all definite or probable MG cases [39,80].

The remaining patients, negative for all available objective studies but still symptomatic, are unlikely to suffer from MG. Notably, such cases that remain undiagnosed after 2 years of observation at a highly specialized MG Clinic may represent over 10% of the referrals [71]. Still, reassessment of some of them, according to clinical judgement, may be indicated, considering the fluctuating course of MG and the possibility for technical or human errors.

## 9. Conclusions

The diagnosis of autoimmune Myasthenia Gravis in the contemporary setting aims at defining the disease subtypes, which inevitably relies on serological methods and makes them the first-line investigation in suspected MG. Re-testing the patient in clinical doubt and using other test modalities makes sense, as cases of “false positives” or delayed seroconversion may be not so rare.

The electrodiagnostic tests have lost some of their significance in the new paradigm, but in seronegative cases, in ocular myasthenia, they may confirm diagnosis and help follow up the patient’s condition objectively.

The pharmacological tests are neglected in some countries with better access to serology and electrodiagnosis, but may still be useful where the appropriate anticholinesterases are available and access to serology and electrodiagnosis is limited. They might replace the electrodiagnosis when unequivocally positive. The icepack is a useful addendum in cases of ocular myasthenia, although in the opinion of this author, the abnormality criteria are somewhat arbitrary.

The possibility to control the great majority of MG cases makes diagnosis of MG gratifying; the advances in treatment that are already around the corner will make knowledge of MG and its timely diagnosis even more responsible and rewarding.

## Figures and Tables

**Figure 1 jcm-10-01736-f001:**
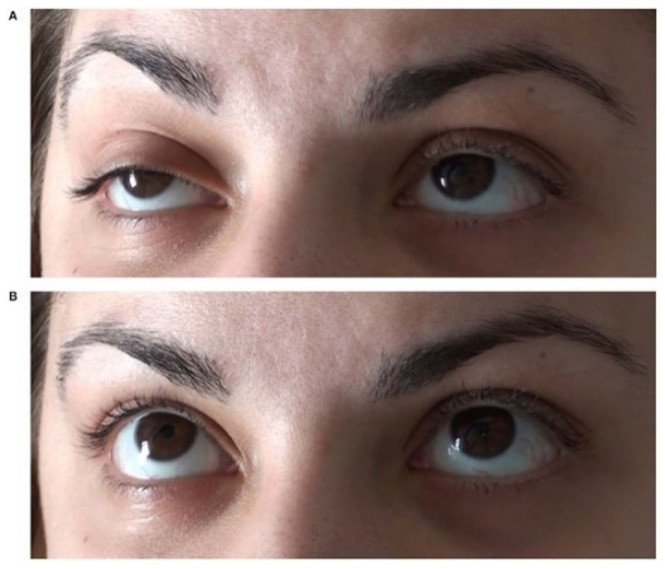
“Mainly right-sided oculomotor paresis with elevation deficit and ptosis (**A**) responded positively to an intravenous test dose of 9 mg edrophonium chloride (**B**), suggesting that double vision was caused by ocular manifestation of myasthenia gravis.” [30].

**Figure 2 jcm-10-01736-f002:**
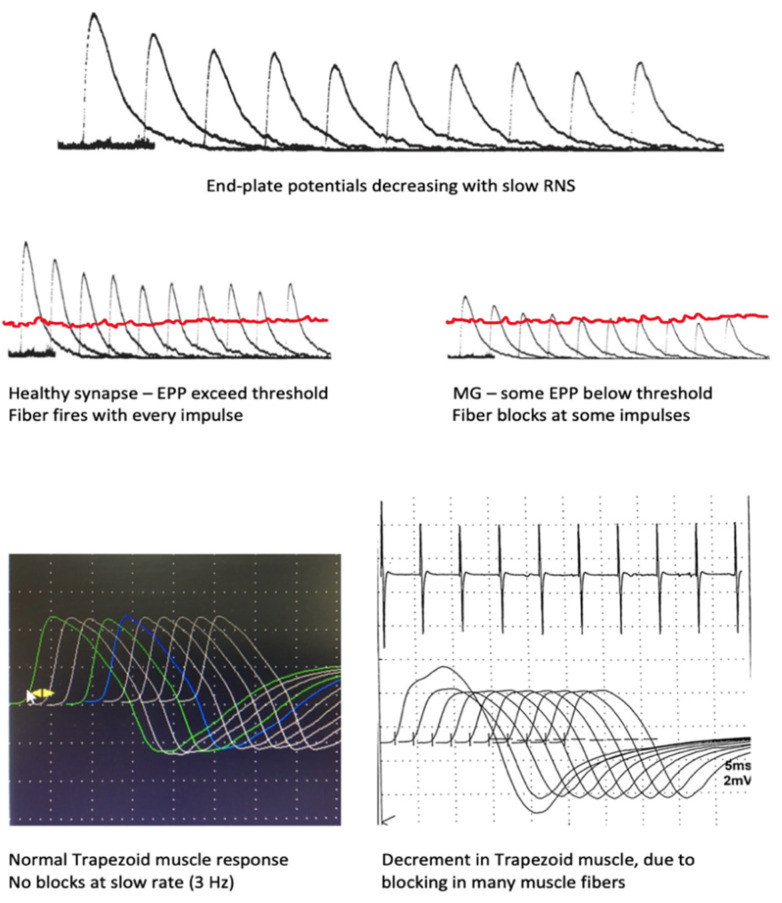
Synaptic events leading to blocking in single muscle fibers and decrement of the compound muscle action potential (CMAP).

**Figure 3 jcm-10-01736-f003:**
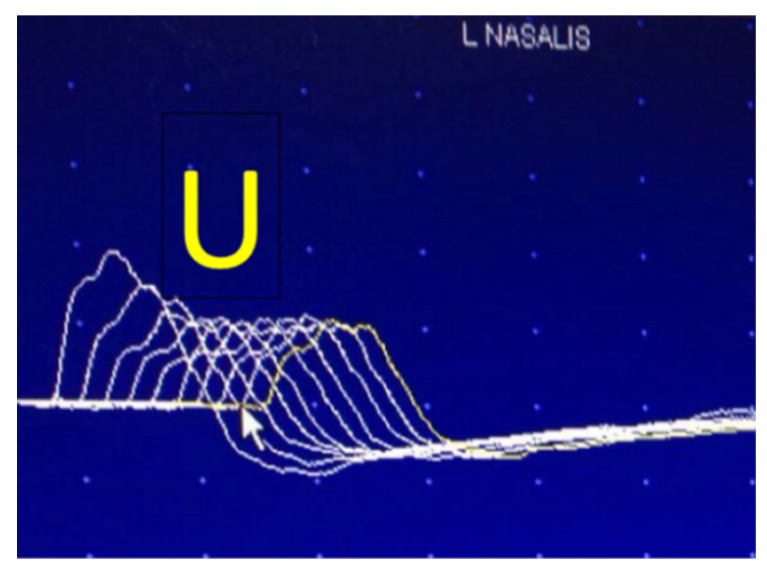
Abnormal decrement in the nasalis muscle in a U-shaped pattern.

**Figure 4 jcm-10-01736-f004:**
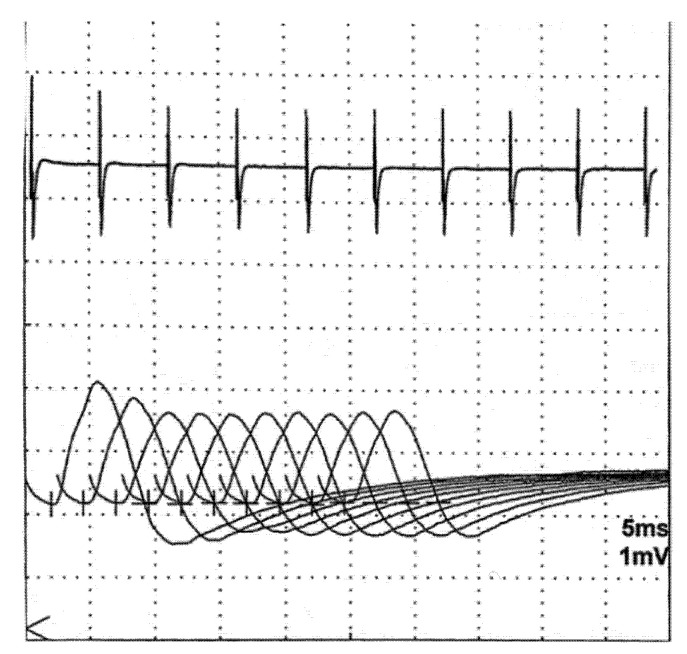
Milder changers in nasalis muscle of another patient, decrement 27.5%.

**Figure 5 jcm-10-01736-f005:**
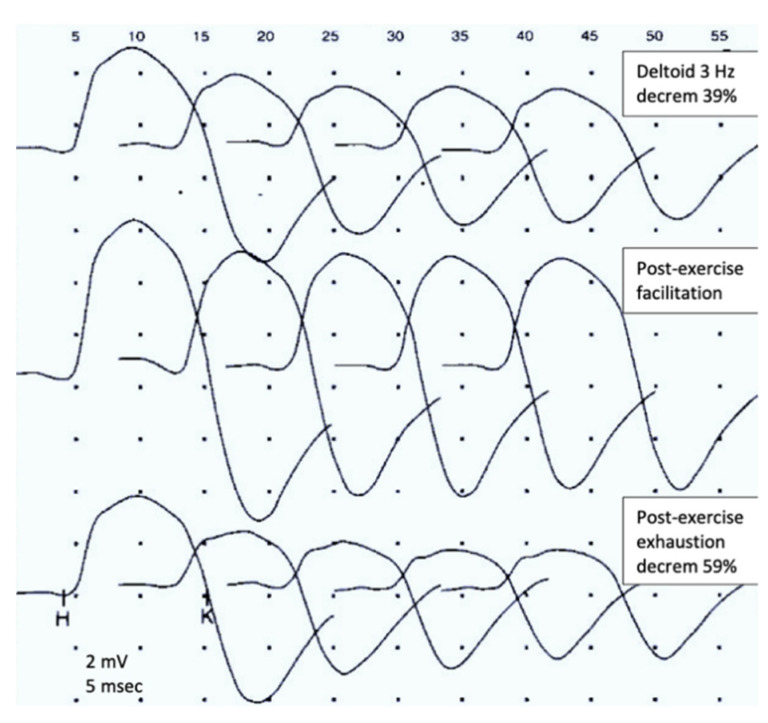
Postexercise facilitation and exhaustion.

**Figure 6 jcm-10-01736-f006:**
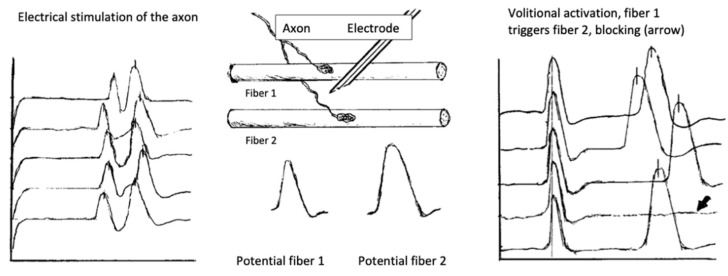
The electrode in position to pick up potentials from Fiber 1 and Fiber 2 at the same time. Electrical stimulation excites both fibers. Some technical aspects (subliminal stimulation, intrusion of “new” fibers with minimal increase in stimulation) make the study less easy than it seems. Volitional jitter is free of false-positives but may be more tricky for the operator and longer, less tolerable, for the patient.

**Figure 7 jcm-10-01736-f007:**
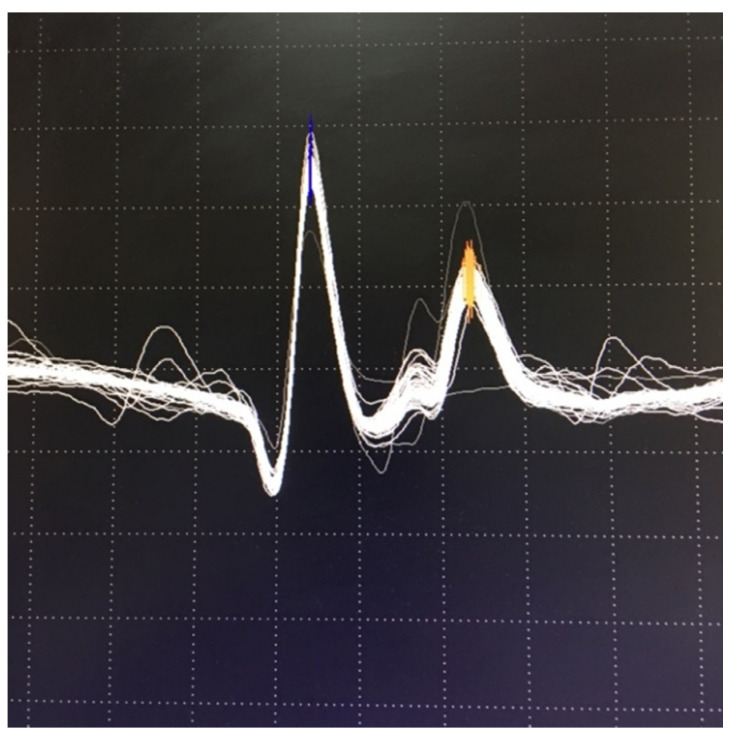
Normal volitional jitter in frontalis muscle: 22 microseconds, grid 0.1 mv by 0.3 msec.

**Figure 8 jcm-10-01736-f008:**
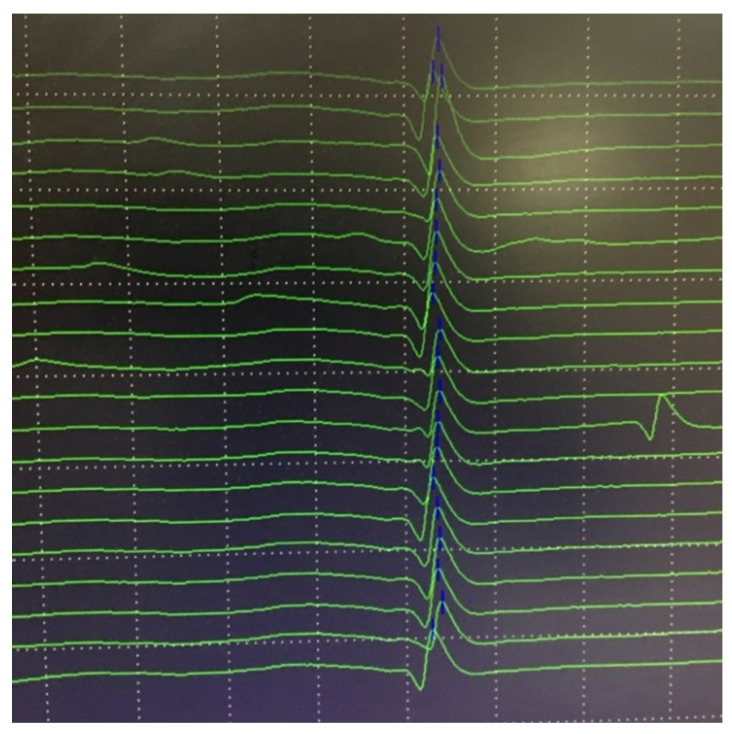
Normal stimulation jitter in orbicularis oculi muscle: 34 microseconds. Grid 1 mv by 0.5 msec.

**Figure 9 jcm-10-01736-f009:**
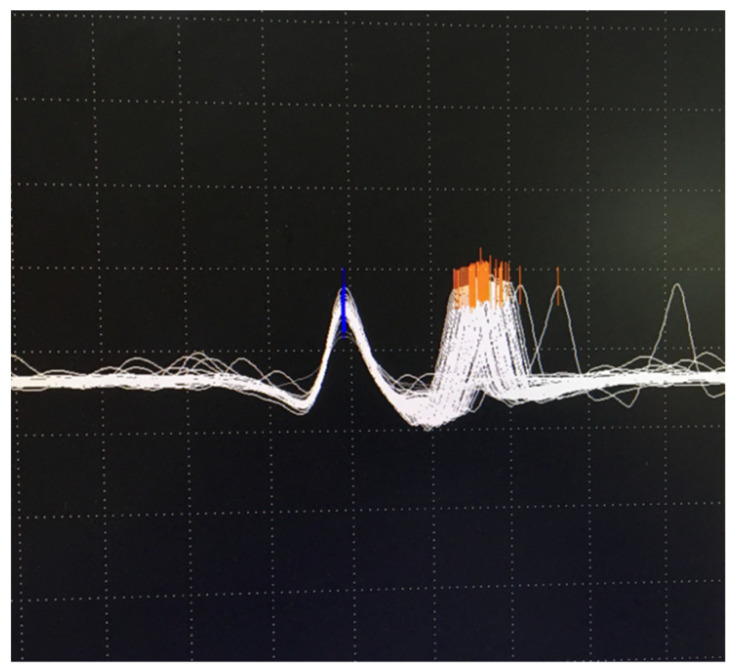
Superimposed traces: abnormal jitter with blocking. Grid 0.1 mV by 0.3 msec.

**Figure 10 jcm-10-01736-f010:**
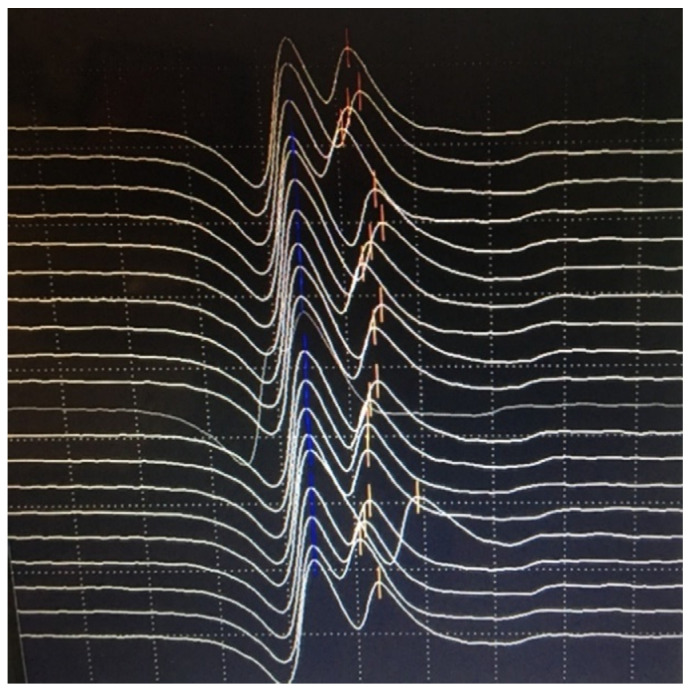
Abnormal jitter in the masseter muscle: cascade mode. Grid 0.5 mV by 0.3 msec.

**Figure 11 jcm-10-01736-f011:**
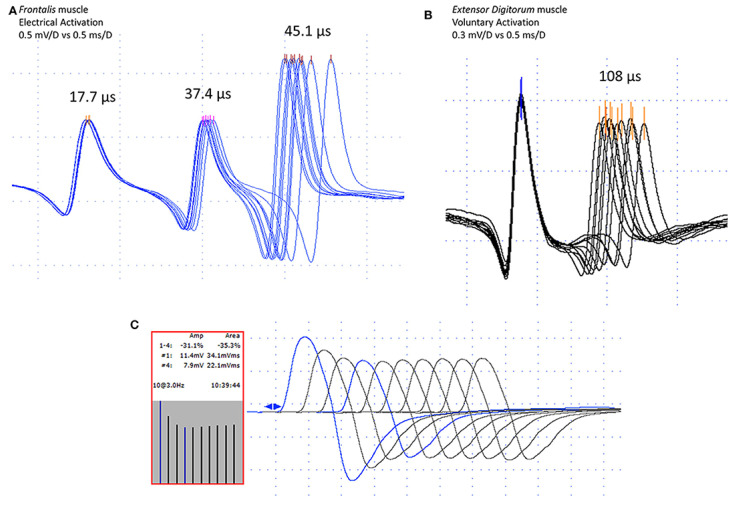
“Ex ungue leonem”: (**A**) flawless recordings of normal and abnormal stimulation jitter, (**B**) abnormal volitional jitter, and (**C**) the paradigmatic U-shaped CMAP decrement, from an article with co-author Professor E. Stalberg. Copyright © 2020 Kouyoumdjian, Paiva, and Stålberg. From: Kouyoumdjian JA, Paiva GP, and Stålberg E. Concentric Needle Jitter in 97 Myasthenia Gravis Patients. Front Neurol. 2020;11:600680. Published 2020 Nov 13. doi:10.3389/fneur.2020.600680, under CC BY license.

## Data Availability

No new data were created or analyzed in this study. Data sharing is not applicable to this article.

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
