# Peer review of "Diagnosis of Myasthenia Gravis"

_jcm, 2021, doi:10.3390/jcm10081736_

Round 1

Reviewer 1 Report

Thank you for asking me to review this article - which I suspect is an invited review. This is well presented and include recent references too (eg: modulating vs binding antibodies). I feel that the article can be shortened by avoiding the extra-long sections about neurophysiology which distracts from the main aim of the paper - i.e.  diagnosis of MG.

Comments noted in order below:

Consistency with abbreviations – ACh vs Ach

Expand MuSK at first mention

Anti-AChR antibodies, Anti-MuSK antibodies etc – probably best to avoid tautology, though am aware that this is used regularly in articles.

NMT – please expand at first mention of neuromuscular transmission

Asystole – spell check – not asystoly

Prefer Ice-pack test to Ice test, which is the standard terminology

Avoid repetition of same information – eg: safety factor (since this is an article on “diagnosis”, probably best to avoid this in the introduction (assuming there would be another articles on pathophysiology).

Figure 2 – would be good to have similar tracings (using same machine and background) for normal vs MG, if practically possible.

Line 297 – even though I understand both these statements – probably best to rephrase these – first line says distal muscles are easier to test and better tolerated and the very next line it mentions that trapezius (the most proximal of the muscles!) is well tolerated and easily performed. The whole paragraph is very confusing – probably highlighting everyone having a “favourite” muscle for various reasons. This can be trimmed significantly.

Line 325 – please rephrase – “more or less painful”?

Line 401 – spacing – “insensitivity”!

Line 433 – clinical and serological

Line 453 – line ends abruptly

Line 454 – I am not sure it is appropriate to say AChR antibodies are sufficient to make a diagnosis – this may be true with a classical presentation, but having seen several patients with non-specific fatigue diagnosed by a single “abnormal” antibody test, it is important that all available tests (SFEMG etc) are utilised to confirm the diagnosis. On the other hand, with a classical clinical presentation (fatiguable ophthalmoplegia and bulbar symptoms), there might not be need for ANY tests before treatments are initiated.

Line 463 - I doubt any myasthenia specialist will agree that a pharmacological test like Edrophonium is clinically meaningful AFTER having had negative antibodies and SFEMG. It might very rarely used as “screening”, but I would advice against widespread use for “confirmation”.

Line 471 – this is very vague – what is the author expecting to find with a brain MRI in a patient with bilateral ophthalmoplegia, bulbar and limb weakness? Agree, in atypical cases – eg: unilateral ptosis/ophthalmoplegia, facial weakness, predominant bulbar presentation etc this will have a role, but a vague statement like “do an MRI head” if antibodies and SFEMG are normal – is not helpful in a review.

I think the Diagnostic Approach needs to be reworded with some practical tips – may be even mentioning that this is the personal approach of the author – using several references here (after having discussed the individual tests in detail) is not very helpful. Can the whole thing be summarised to a Box/Flowchart?

Conclusion – again the author relies on antibodies when it is only positive in 50% of ocular MG, whereas the neurophysiology has well over 95% sensitivity. Also, paradoxically the author using 230 lines to describe neurophysiology and only has 75 lines to describe several antibodies – making me wonder whether the author is mainly a neurophysiologist!

There is no detailed mention about congenital myasthenia anywhere in the article, yet the author decided to bring this to the conclusion, which can be avoided please.

Like the Diagnostic Approach – the conclusion needs to be reworded substantially and avoid repeating what was mentioned literally two paragraphs earlier.

In summary, brevity would be good – especially for the neurophysiology section (which in my opinion is disproportionately long) and also in the last two sections.

Author Response

Dear respected Reviewer, I am very happy with  your opinion on my work. I have corrected the technical omissions noted

Sincerely

Dr Rossen

Reviewer 2 Report

The article is a comprehensive overview of the current diagnostic tests for the MG and follow-up of MG.

Strength

The author included sufficient information on the topic, and the information is relevant. The article is helpful for the clinicians as it provided the pros and cons of the approaches to diagnose MG reliably.

Limitations

The citation style of No. 68 reference should be corrected.

Author Response

Dear Respected Reviewer,

Thank you very much for the thorough and critical remarks and suggestions, I have tried to redact the paper using them as fully as possible:
